# OpenReview forum: "Beyond Next-Token Alignment: Distilling Multimodal Large Language Models via Token Interactions"
_ICLR.cc/2026/Conference — Submitted to ICLR 2026_

### Official Review · Reviewer_dQg7 · 2025-10-27

**Soundness:** 2
**Presentation:** 3
**Contribution:** 2
**Rating:** 4
**Confidence:** 3

**Summary:**

This paper introduces Align-TI, a token-level knowledge distillation framework for compressing multimodal large language models (MLLMs). It emphasizes token interactions through two components: Instruction-aware Vision Alignment (IVA), aligning visual tokens with instruction-relevant regions, and Transition Probability Alignment (TPA), transferring token-to-token generation dynamics. Experiments show Align-TI effectively distills MLLMs, offering a fine-grained approach to multimodal knowledge transfer.

**Strengths:**

1. This paper carefully analyzes the current research landscape of vision–instruction token interactions and intra-response token interactions, and on that basis proposes Instruction-aware Vision Alignment (IVA) and Transition Probability Alignment (TPA) to address these issues.
2. Experimental results demonstrate the effectiveness of Align-TI, showing that with fewer parameters, it still surpasses larger models like LLaVA-1.5-7B by 7.0%, offering a new perspective for parameter-efficient MLLM distillation.
3. Overall, the paper is well-written and easy to understand.

**Weaknesses:**

1. **Motivation:** The motivation of this paper is based on the premise that the current “static next-token alignment” has fundamental limitations, whereas “dynamic token interactions” capture critical capabilities for MLLM understanding and generation. However, this premise lacks both conceptual clarity and empirical support. First, the paper does not provide operational definitions for “static” and “dynamic” making it difficult to understand why existing methods (e.g., Vanilla KD) are classified as “static”. Second, the paper merely claims that static alignment is insufficient without offering concrete evidence or task-based examples.
2. **Novelty:** 1) "Align-KD: Distilling Cross-Modal Alignment Knowledge for Mobile Vision-Language Models" has already proposed enhancing distillation via cross-modal alignment and applied it before the input is fed into the model. Your proposed IVA module is quite similar to Align-KD, by leveraging IRS, you apply this mechanism across different layers. Could you clarify the core design differences between the two? 2) The Instruction-Relevant Score (IRS), which serves as the basis for all subsequent experiments, lacks sufficient justification. 3) TPA increases computational complexity and it is unclear whether this increase is justified by the performance gains.
3. **Experiments & Results：** 1) Unclear ablation explanation: In Table3, IVA alone brings a 0.8% improvement, but when introduced after TPA, the gain drops to only 0.3%. Do the two modules have overlapping or conflicting functionalities? 2) Why does Table 4 not include a comparison with the distilled model?
4. **Format:** 1) The “6% / 4.6% improvement over Vanilla KD” shown in Figure 1 cannot be clearly verified in Table 1, and it is unclear which method is referred to as Vanilla KD. 2) Line 360: “GKD” is unclear—please specify whose abbreviation it is.

**Questions:**

see weakness

---

> ### Author Response · Authors · 2025-11-21
> **Response to Reviewer dQg7 (1/5)**
>
> We sincerely appreciate your constructive comments and valuable suggestions. Thank you for the time and effort you have devoted to assessing our manuscript and helping us improve its quality. Our detailed responses to your concerns are provided below.
>
> ---
>
> > **Q1.1: “Static next-token alignment” v.s. “dynamic token interactions”**
>
> We thank the reviewer for the insightful comments.
>
> We use "static" to describe the traditional next-token alignment process, in which prefix sequences are sourced from a fixed, pre-collected dataset. The training objective relies on **off-policy** sampled token sequences. This means that during training, the model is consistently optimized using the same static context distribution, without dynamically adjusting to its own generated content. A related description can be found in Figure 2 (Right) and lines 100–102 of our paper.
>
> In contrast, "dynamic" refers to the process where interactions between tokens change in real-time as generation progresses during model inference, which is an **on-policy** sequential decision-making process. In this paper, we further divide this dynamic interaction into two stages (as visualized in Figure 2):
>
> 1.  **Prefilling Stage (Figure 2, Left):** The model dynamically adjusts its attention to Visual Tokens based on the Instruction Tokens to extract key visual information.
> 2.  **Decoding Stage (Figure 2, Right):** The model needs to continuously predict the next token based on the prefix sequence it has generated. This prediction, based on its own output, continuously changes the state distribution and token dependency structure.
>
> ---
>
> > **Q1.2:  Static alignment is insufficient.**
>
> We thank the reviewer for this valuable comment. Our paper provides both conceptual analysis and empirical evidence to substantiate the insufficiency of static alignment.
>
> **Conceptually**, in the Figure 2 (Right) and its accompanying description, we discuss in detail why static next-token alignment is inadequate. Specifically, this static alignment method creates a significant gap between training and testing. During training, the prefix sequences are entirely data-conditioned (i.e., derived from the ground-truth data). In contrast, during inference, the prefix sequences are self-conditioned, as they are generated autoregressively by the model itself. This discrepancy between the distributions of data-conditioned and self-conditioned prefixes causes a mismatch that is progressively amplified during inference, leading to autoregressive accumulated error.
>
> **Empirically**, to further substantiate this point, we have plotted the accumulated error over generation steps for a model distilled with Vanilla KD, comparing its performance during training time and test time (bottom of the right panel in Figure 2). The plot clearly shows that this gap between training and testing behavior widens as the number of generation steps increases, providing concrete evidence. The detailed calculation method for this figure is provided in Appendix B.4.1.

---

> ### Author Response · Authors · 2025-11-21
> **Response to Reviewer dQg7 (2/5)**
>
> > **Q2.1: Comparison with Align-KD**
>
> We thank the reviewer for this insightful comment. While both our proposed IVA module and Align-KD leverage instruction-to-vision attention maps, we would like to highlight that they are fundamentally different in their motivation and methodological design.
>
> **1. Motivation:**
>
> -   Align-KD: The primary motivation is to train the **adaptor** to **distill the teacher model's representation of visual tokens.**
> -   IVA: The primary motivation is to train the **LLM** to **learn and mimic how the teacher model processes and extracts visual information.**
>
> **2. Methodological Design:**
>
> -   **Training Object:** Align-KD focuses on training the **adaptor**, while IVA mainly focus on training the **LLM**.
>
> -   **Source of Attention Maps:** Align-KD exclusively uses the attention map from the **first layer**. In contrast, IVA employs the IRS to identify the **layer with the most instruction-relevant attention map.** Additionally, our analysis (as shown in Figure 12 and Appendix C.2) reveals that the first layer's attention map is largely instruction-agnostic, exhibiting similar patterns across different instructions.
>
> -   **Distillation Target:** Align-KD adopts the visual tokens before they are fed into the LLM, while IVA adopts the final visual tokens produced by the LLM’s output head.
>
> -   **Loss Function:** Align-KD performs distillation in a **continuous feature space** by minimizing the Mean Squared Error (MSE) loss on visual token representations. In contrast, IVA operates in the **discrete vocabulary space** and minimizes the KL Divergence loss.
>
> ---
>
> > **Q2.2: Lacks sufficient justification of IRS.**
>
> We thank the reviewer for this insightful comment. We respectfully point out that we have dedicated a significant portion of our manuscript to justifying the IRS. We summarize the key justifications as follows:
>
> 1.  **Motivation for Introducing IRS:** We explain the rationale for introducing IRS in both the **Introduction (Lines 111-114)** and **Method (Lines 220-228)** sections. Our proposed IVA aims to enable the student model to mimic the teacher’s ability to extract instruction-relevant visual information by aligning on salient visual regions. These salient regions are identified using instruction-to-vision attention maps. However, the characteristics of these attention maps vary significantly across different layers. Therefore, we proposed the IRS to quantitatively measure the relevance of an attention map to the given instruction, allowing us to select the most instruction-relevant attention map for alignment. Additionally, this approach was endorsed by **Reviewer vd3n**, who noted, *"The IRS metric is a good method for selecting the most relevant layer, rather than relying on manual design."*
>
> 2.  **Principle and Definition of IRS:** The formal definition of IRS is provided in **Lines 210-215**, followed by a remark in **Lines 216-217**. The core principle of IRS is intuitive: if a layer's attention maps are highly similar across different instructions, that layer is likely instruction-agnostic. Conversely, a layer that produces distinct attention maps for different instructions is instruction-relevant. We use cosine similarity to quantify this relationship.
>
> 3.  **Generality of IRS:** In **Appendix C.1**, we demonstrate that the observed behavior of IRS (first increasing, then decreasing across layers) is not unique to the Qwen model family. We show that LLaMA-based Multimodal Large Language Models (MLLMs) exhibit a similar pattern. This suggests that IRS is a generalizable metric applicable across different model architectures.
>
> 4.  **Analysis of IRS Behavior:** We provide a detailed discussion of how IRS varies across layers in **Lines 200-237**, complemented by **Figure 4**. To offer deeper insight, we also include a visual analysis of this behavior in **Appendix C.2**, illustrating how attention maps change in correspondence with the IRS values.
>
> 5.  **Computational Details:** To ensure reproducibility, we have provided the specific implementation details for calculating the IRS in **Appendix B.5**.
>
> We hope this summary clarifies the extensive justification provided for the IRS in our paper and addresses the reviewer's concern.

---

> ### Author Response · Authors · 2025-11-21
> **Response to Reviewer dQg7 (3/5)**
>
> > **Q2.3: About computational complexity.**
>
> We thank the reviewer for this insightful comment. We have discussed the performance improvements of TPA (in Table 4) and the additional training overhead (in Table 6) in our manuscript.
>
> To further clarify this point, we would like to reiterate the results here. As shown in the table below, TPA consistently outperforms Vanilla KD across all benchmarks. The improvements are particularly significant on the MME and MMB benchmarks, with gains of 3.1 and 2.6 points, respectively. The cost for this improvement is a 1.4x increase in training time compared to Vanilla KD, while the GPU memory usage remains almost the same. Given the substantial performance improvements achieved, this additional one-time training overhead is an acceptable trade-off.
>
> | Method     | Training Time (H) | Memory (GiB) | GQA  | SQA  | TextVQA | POPE | MME  | MMB  | AVG  |
> | ---------- | :-----------------: | :------------: | ---- | ---- | :-------: | ---- | ---- | ---- | ---- |
> | Vanilla KD | 355               | 70.6         | 59.3 | 59.7 | 59.2    | 86.2 | 65.0 | 60.4 | 65.0 |
> | TPA        | 504               | 75.3         | 60.3 | 61.0 | 59.6    | 86.5 | 68.1 | 63.0 | 66.4 |

---

> ### Author Response · Authors · 2025-11-21
> **Response to Reviewer dQg7 (4/5)**
>
> > **Q3.1: The performance gain from IVA is not significant.**
>
> We thank the reviewer for this insightful question.
>
> First, we would like to clarify that IVA yields a more substantial improvement in other settings. Specifically, when applied on top of SFT, IVA achieves an average improvement of +0.8 across multiple benchmarks, which indicates that IVA is indeed effective as an independent component.
>
> Second, for the observed smaller gain when combining IVA with TPA, we have discussed this phenomenon in Lines 395–399. The reasons can be summarized as follows:
>
> - **Principle Perspective (Lines 395-397):** The core objective of MLLM KD is to align the student model's responses with the teacher model. TPA **directly** optimizes response-level alignment. In contrast, IVA operates **indirectly** by guiding the student to imitate the teacher's processing of visual tokens, thereby refining hidden representations for better response generation. When combined, **TPA's direct alignment leaves limited room for IVA's indirect optimization to yield substantial gains.**
>
> - **Overhead Perspective (Line 397-399):** While IVA's performance gain over TPA is limited, it can be integrated with TPA at **near-zero cost**. As shown below:
>
>   | Method  | Training Time (H) | GQA  | SQA  | TextVQA | POPE | MME  | MMB  | AVG         |
>   | ------- | ----------------- | ---- | ---- | ------- | ---- | ---- | ---- | ----------- |
>   | SFT     | 242               | 57.6 | 59.0 | 59.1    | 86.2 | 66.1 | 57.9 | 64.3        |
>   | TPA     | 503 **(x2.1)**    | 60.3 | 61.0 | 59.6    | 86.5 | 68.1 | 63.0 | 66.4 (+2.1) |
>   | TPA+IVA | 509 **(x2.1)**    | 60.4 | 60.7 | 59.9    | 86.8 | 68.9 | 63.2 | 66.7 (+2.4) |
>
> ---
>
> > **Q3.2: Why does Table 4 not include a comparison with the distilled model?**
>
> We thank the reviewer for this valuable comment.
>
> The primary purpose of distillation is to train smaller, more efficient models. In this context, the goal of Table 4 (now Table 5 in the revised manuscript) is to showcase the significant efficiency gains of our distilled small model. Therefore, we compared our model's inference efficiency and performance against a much larger 7B model.
>
> Including other distilled models in this table would be less meaningful. Since other distillation methods also produce models with the same parameter size as ours, their inference efficiency would be virtually identical. The key differentiator between our method and other distilled small models lies in the performance, which we have already comprehensively compared in Table 1.
>
> Table 5 (inference efficiency) and Table 6 (training efficiency) together form the complete efficiency evaluation of our proposed method.

---

> ### Author Response · Authors · 2025-11-21
> **Response to Reviewer dQg7 (5/5)**
>
> > **Q4.1: Figure 1 cannot be clearly verified in Table 1.**
>
> We thank the reviewer for this comment. In Table 1, the reported results correspond to **Figure 1 (Left)**, where we benchmark our models against state-of-the-art MLLMs, including both models trained from scratch and those derived via distillation. The “6% / 4.6% improvement” mentioned in **Figure 1 (Right)**, however, comes from our ablation study, and the corresponding numerical results can be found in Tables 3 (Line 369) and 4 (Line 376). Furthermore, we have added a new table in the appendix that contains all the numerical data plotted in Figure 1 (Right).
>
> ---
>
> > **Q4.2:  “GKD” is unclear—please specify whose abbreviation it is.**
>
> We thank the reviewer for their meticulous review. GKD is the abbreviation for Generalized Knowledge Distillation [1], a distillation method for LLMs. We apologize for omitting the citation in the main text. We have now added the reference on Line 360. We sincerely thank the reviewer for pointing this out.
>
> [1] Agarwal, Rishabh, et al. "On-policy distillation of language models: Learning from self-generated mistakes." ICLR. 2024.
>
> ---
>
> We would be more than happy to discuss any further questions!

---

> ### Comment · Reviewer_dQg7 · 2025-11-22
> **Response to the rebuttal**
>
> I thank the authors for the rebuttal, which has addressed some of my questions regarding training overhead, IRS, and "static versus dynamic" understanding. I appreciate the model's performance; however, I believe it still does not sufficiently demonstrate novelty compared to related work. For example, IVA is quite similar to Align-KD (whereas IRS is relatively more innovative), and there are still several aspects of TPA's implementation that warrant further discussion. Therefore, my concerns remain, and I will maintain my original score.

---

> ### Author Response · Authors · 2025-11-22
> **Response to Reviewer dQg7 (1/3)**
>
> We sincerely thank the reviewer for their continued engagement and prompt feedback. We are encouraged that our previous rebuttal has resolved some of your concerns. We understand that your primary remaining concern is about the novelty of our method, particularly when comparing our IVA module to Align-KD.
>
> However, we respectfully argue that viewing IVA in isolation and comparing it to Align-KD does not fully capture the novelty of our work, and the comparison is even unfair for us. **Our main contribution is the proposal of a unified MLLM KD framework from the perspective of Token Interactions**. This framework includes **not only IVA**, which addresses Vision-Instruction Token Interactions, **but also TPA**, which tackles Intra-Response Token Interactions. As discussed in Figure 2, these two interaction modes present two challenges: IVA was developed to address the long-tail distribution and dynamic attention characteristics of visual tokens by aligning on salient regions, while TPA was designed to mitigate the training-test gap observed in response token generation.
>
> Our detailed responses to your concerns are provided below.

---

> ### Author Response · Authors · 2025-11-22
> **Response to Reviewer dQg7 (2/3)**
>
> > **Q1: The difference between Align-KD and IVA in our framework.**
>
> We first thank you for acknowledging the innovation of the IRS method used in our IVA module, a point also appreciated by reviewer vd3n. **Your novelty concern seems to focus on IVA, like Align-KD, utilizes the importance of visual tokens derived from attention maps.**
>
> We acknowledge this similarity, but we must emphasize that leveraging visual token importance is not a new concept. Prior works on MLLM visual token pruning [1-2] have used this idea long before Align-KD. **The novelty of a method should be judged by the specific problem it solves and how it utilizes this information, rather than by the mere use of a general principle.**
>
> **(1)** In this regard, the way IVA and Align-KD utilize visual token importance is fundamentally different:
>
> -  **IVA** enables the student model to learn how the teacher model processes and handles visual information. **The attention map is introduced to guide the student's learning on the regions that the teacher model focuses on.** The underlying rationale is that visual tokens exhibit a long-tail distribution, and we use this importance weighting to prevent the student model, which has limited capacity, from focusing its attention on redundant visual tokens.
> -  **Align-KD** aligns the student's visual token representations with the teacher's. **The attention map is introduced to "help the projector get more exposure to cross-modal information."**
>
> **(2)** Furthermore, although both methods use attention maps, the nature of the attention maps employed by IVA and Align-KD is significantly different.
>
> - **IVA** requires aligning on **instruction-relevant** regions, so we use our proposed IRS to detect the most instruction-relevant attention map.
> - **Align-KD** uses the attention map from the **first layer**, based on their experimental finding that **this layer "brings the largest shift on the features."** It does not emphasize instruction relevance. In fact, our own experiments with IRS show that the first layer's instruction relevance is very weak, less than 5% for both Qwen-based and LLaMA-based MLLMs.
>
> **(3)** In addition, we would like to summarize the differences between Align-KD and IVA, previously described in our response to Q2.1:
>
> |                          |                      Align-KD                      |                          IVA (Ours)                          |
> | ----------------------- | ----------------------------------------------- | ---------------------------------------------------------- |
> | Motivation               |     Distill the visual tokens representation.      | Mimic how the teacher model processes and extracts visual information. |
> | Main Training Object     |                      Adaptor                       |                             LLM                              |
> | Source of Attention Maps |                    First layer.                    | Layer with the most instruction-relevant attention map (found by IRS). |
> | Distillation Target      | Vision token features **before** LLM (continuous). |   Vision token importance scores **after** LLM (discrete).   |
> | Loss Function            |           MSE (for continuous features).           |        KLD (for discrete probability distributions).         |
>
> Given these fundamental differences in motivation, mechanism, and implementation, we respectfully argue that IVA is distinct from and more sophisticated than Align-KD. We are also encouraged that Reviewers SRQm, vd3n, and Kx6f have recognized the novelty and insights of our IVA module and the overall framework.
>
> We hope this detailed explanation addresses your concerns. We are more than happy to engage in further discussion on this topic. Thank you again for your valuable time and feedback.
>
> [1] Huang, Kai, et al. "Ivtp: Instruction-guided visual token pruning for large vision-language models." ECCV, 2024.
>
> [2] Ye, Xubing, et al. "Atp-llava: Adaptive token pruning for large vision language models." CVPR, 2025.

---

> ### Author Response · Authors · 2025-11-22
> **Response to Reviewer dQg7 (3/3)**
>
> > **Q2: TPA Implementation in Align-TI Warranting Further Discussion.**
>
> We thank the reviewer for this comment.  As this concern was not mentioned in the initial review, we are not entirely certain which "several aspects of TPA's implementation" the reviewer believes warrant further discussion. To proactively address this, we would like to briefly summarize the description of our TPA implementation already present in the manuscript:
>
> 1. **Objective Derivation (Lines 245-264):** In conjunction with Figure 3, we describe that our objective is to align not only the immediate next-token distribution but also the subsequent transition probabilities. We further discuss the rationale behind our objective, including (1) the necessity of sampling, and (2) the reason for sampling from the student model.
> 2. **Computational Method (Lines 229-305):** We explain how we use the Monte Carlo method to estimate the expectation term within our objective function.
> 3. **Parallel Acceleration (Lines 306-317):** We detail how the Monte Carlo computation can be parallelized to improve training efficiency.
>
> To further enhance clarity, we will add a new **Appendix B.6 containing a detailed algorithmic flowchart for TPA.** This will provide a cohesive, step-by-step walkthrough of the TPA implementation process. We believe this addition will be very helpful for readers, and we are grateful to the reviewer for prompting us to reflect on this. We expect to complete this addition within 1-2 days.
>
> Furthermore, regarding novelty, we would like to gently point out that Reviewers Kx6f and SRQm both affirmed our work's novelty. Reviewer Kx6f even stated that our TPA module explores an "uncharted problem in the multimodal domain." We hope this, along with the planned improvements to the TPA description, can help alleviate the reviewer's concerns.
>
> If the reviewer could specify any remaining unclear aspects of TPA, we would be more than happy to discuss them in detail. Thank you again for your time and effort in reviewing our work.

---

> ### Author Response · Authors · 2025-11-24
> **Official Comment by Authors**
>
> Dear Reviewer dQg7,
>
> Thank you once again for your time in reviewing our rebuttal. We have included Appendix B.6, which provides a detailed algorithmic flowchart of the TPA implementation. We hope this addresses your concerns regarding the implementation of TPA.
>
> We would be very grateful for a corresponding update of the score if you feel that our revisions and rebuttal have sufficiently addressed your concerns. Thank you again for your helpful comments, and we are happy to clarify any further questions you may have about the paper.
>
> Sincerely, Paper #9359 Authors

---

> ### Comment · Reviewer_dQg7 · 2025-11-26
> **Response to the rebuttal**
>
> Thank you for the response. I have read the newly added Section B.6, which alleviates my concerns about TPA to some extent. However, the paper states that the core contribution of TPA is to align token interactions in order to model richer “generation logic.” I notice that the proposed formulation adopts a one-step Markov transition. This raises a concern: does relying on one-step transitions introduce potential limitations when handling long-range dependencies? I understand that existing KD methods are also generally based on next-token supervision, but since TPA specifically emphasizes modeling token-to-token interactions, it becomes more important to clarify whether this one-step assumption is sufficient for capturing longer-range dependencies (e.g., referent resolution, cross-sentence coherence, global semantic consistency). If the authors could further explain or discuss this point, it would greatly strengthen the paper’s clarity and contribution. I am willing to raise my score based on a clearer justification.

---

> ### Author Response · Authors · 2025-11-27
> **Response to Reviewer dQg7 (1/2)**
>
> We sincerely thank the reviewer for reading our rebuttal and for the time and effort dedicated to reviewing our work. We are also grateful for your consideration in raising the score. We appreciate this insightful question, which has prompted us to rethink the underlying mechanics of our approach.
>
> ---
>
> We understand the reviewer's concern is whether the one-step Markov transition assumption in our TPA introduces limitations in capturing long-range dependencies. In fact, the one-step transition in TPA is a **conditional** one-step Markov transition. **It is conditioned on the entire given prefix, which includes the source input $\boldsymbol{x}$ and the previously generated tokens $\boldsymbol{y} _ {<k}^\mathcal{D}$.** ***[Note]*** This means that the transition probability alignment is calculated with the full context of the preceding sequence. Therefore, the ability to model long-range dependencies is **fully preserved and not compromised by TPA**, and TPA does not introduce the potential limitations on handling long-range dependencies that the reviewer mentioned.
>
> ***[Note]*** In our original manuscript, we omitted the shared prefix $\boldsymbol{x}, \boldsymbol{y}_{<k}^\mathcal{D}$ in Equation 5 since it is too long, and to better highlight the contrast between Initial State Alignment and Transition Probability Alignment. We acknowledge that this simplification may have led to the misunderstanding. Therefore, we have revised Equation 5 to explicitly include this conditioning, making the formulation clearer.

---

> ### Author Response · Authors · 2025-11-27
> **Response to Reviewer dQg7 (2/2)**
>
> **Beyond preserving this capability, TPA promotes the student's ability to learn the teacher's long-range dependencies.** To explain this, we consider the alignment from the perspective of the full sequence joint space. The ideal distillation goal is to minimize the KL divergence between the teacher's and student's distributions over the entire sequence: $\mathop{\min} _ {\theta} \mathbb{E} _ {\boldsymbol{x} \sim \mathcal{D} _ x} \left[ D_{\text{KL}}\left(P _ T \parallel P_S^{\theta}\right)(\boldsymbol{y} _ {1:L} \mid \boldsymbol{x}) \right].$
>
> If this objective is better optimized, it means the student has captured more of the teacher's *continuous generation logic*, thereby improving its ability to handle long-range dependencies and transcending the limitations of simple token-level mimicry. Our perspective is that TPA facilitates the optimization of this sequence-level objective. We provide both theoretical and experimental evidence to support this:
>
> - **Theoretically.** Strictly optimizing this objective necessitates alignment within a joint probability space of complexity $O(|V|^L)$. In practice, this space is computationally intractable and highly sparse, with numerous combinations being semantically meaningless or contextually irrelevant. As discussed in Appendix D.1, Vanilla KD simplifies this objective by performing alignment in an $O(|V|)$ space. In contrast, TPA operates within an $O(|V|^2)$ space. This implies that TPA exposes the student to richer structural patterns and transition dynamics during training. Such alignment provides a superior approximation of the full sequence distribution compared to Vanilla KD, thereby facilitating the optimization of this objective.
> - **Empirically.** A primary motivation for TPA is to mitigate the Exposure Bias arising from the training-test distribution shift. We employ the Excess Accumulated Error metric ($\mathrm{\%ExAccErr}$) to quantify the severity of this bias. Fig.7 illustrates the trajectory of $\mathrm{\%ExAccErr}$ as the number of generation steps increases. The results indicate that the model distilled via TPA maintains a remarkably low error rate (ranging between 0 and 10\%), which is significantly lower than the approximately 30\% observed with Vanilla KD. This substantial reduction suggests that the cumulative error between the student and teacher is effectively suppressed during autoregressive generation. Consequently, the sequences generated by the student exhibit higher fidelity to the teacher's distribution, empirically confirming this objective is better optimized under TPA.
>
> **In summary, TPA not only preserves the language model's inherent mechanism for capturing long-range dependencies, but it also actively assists the student model in learning the teacher's continuous generation patterns, thereby promoting a stronger alignment between them.** We are very grateful for the reviewer's insightful feedback, which has helped us to reflect on and clarify this point. **We have added a new discussion in Appendix D.3, "Discussion on Impact of TPA on Sequence-level Alignment," to incorporate these into the paper.**
>
> ---
>
> Thank you again for your valuable contribution to improving our work. We are happy to discuss any further concerns.

---

### Official Review · Reviewer_fYcx · 2025-10-29

**Soundness:** 2
**Presentation:** 3
**Contribution:** 2
**Rating:** 4
**Confidence:** 4

**Summary:**

This paper aims to compress the model size of MLLMs, using knowledge distillation techniques. Beyond traditional KD that focuses on next-token alignment, authors turn to achieve KD via token interactions and propose Align-TI, which involves two main components on vision and transition probability alignment. This paper explicitly discusses vision-instruction token interactions and intra-response token interactions, rather than directly aligning the next-token probabilities.
The vision-instruction token is weighted by aggregating the attention weights. The transition probability alignment is achieved by sampling from the student model and aligning the transition probabilities between teacher and student. Extensive experiments on various benchmarks demonstrate the effectiveness of the proposed method.

**Strengths:**

1. The idea of aligning token interactions instead of next-token probabilities is straightforward and motivated with illustrative examples, as shown in Figure 2, making it easy to understand the intuition behind the method.
2. The design of the proposed method is detailed and easy to follow.
2. Extensive experiments are conducted, demonstrating the effectiveness of the proposed method.

**Weaknesses:**

1. The theoretical analysis is highlighted as a contribution in the introduction, but it is not well elaborated and limited in the main text, which seems to contribute less to the method.
1. This paper only focuses on the projector-based multimodal language models, ignoring other research lines, like the unified multimodal model.
2. No code release is mentioned, which hinders reproducibility and further research on related topics. The parallelized calculation seems non-trivial to implement. And there are many hyper-parameters to sample tokens.
3. There are some differences between Equations 2 and 5 on the vanilla KD loss, which may confuse readers. Please make it consistent.
4. The performance of the teacher model is not reported in the main text (I found it in the appendix, should be transferred to the main text) for a comprehensive comparison. Since the teacher model is already powerful, it is important to see how much performance drop occurs after distillation rather than comparing with other weaker baselines.
5. The efficiency analysis seems redundant and unfair, since a smaller model is naturally more efficient than larger models. Do you want to highlight the efficiency of your distillation process (e.g., Parallelized Calculation) compared to the models of same size? If yes, please clarify it.

**Questions:**

Vision-instruction token interaction alignment is quite straightforward. However, the transition probability alignment seems a little bit tricky. It seems you actually want to align next two tokens; however, at the k-th step, you have not determined $y_k$, so you sample $y_k$ from the student model, and then align the transition probability from $y_k$ to $y_{k+1}$ between teacher and student. So I have the following questions:
1. Can you implement a direct alignment of the next two tokens, i.e., align $P(y_{k+1}|y_k, x)$ between teacher and student, where $y_k$ and $y_{k+1}$ are both from the ground-truth response? If yes, how does it perform compared to your proposed method?
2. Can you sample from the teacher model instead of student model to compute the transition probability? How does it perform?
3. Can you cancel the vanilla KD loss, only using the proposed two alignment losses? How does it perform?

For the experiments, I have the following questions need to be clarified:
1. In Tables 2 and 3, do you only report the performance on adapting different methods to LLMs, not MLLMs? If yes, how do you implement IVA (the vision token alignment) on LLMs? Can you clarify it?
2. Following up on the above question, in Table 3, why does Align-TI (w/ IVA and TPA) have the same performance as in Table 2? Table 3 should be an ablation study for MLLMs. In my understanding, it should have the same performance as Table 1 (the overall performance comparison in MLLM setting).
3. In Table 6, selecting layer 0 also achieves a good performance, which seems to contradict your claim. This can help understand the importance of IRS.
4. Also, the same question, why is the performance in Tables 6, 7, and 8 different from Tables 2 and 3? Are they all conducted on MLLMs or LLMs? Please clarify it.

---

> ### Author Response · Authors · 2025-11-21
> **Response to Reviewer fYcx (1/4)**
>
> We sincerely appreciate your constructive comments and valuable suggestions. Thank you for the time and effort you have devoted to assessing our manuscript and helping us improve its quality. Our detailed responses to your concerns are provided below.
>
> ---
>
> > **Q1: Theoretical analysis insufficiently elaborated.**
>
> We thank the reviewer for this valuable comment. We would like to clarify a potential misunderstanding regarding our claim of "theoretical analysis" as a contribution.
>
> In the introduction, we state: *"Moreover, both theoretical analysis and experimental evidence demonstrate that TPA helps mitigate the teacher-student autoregressive generation discrepancy at test time."* **The "theoretical analysis" mentioned here refers specifically to our analysis of how our proposed TPA method works to alleviate the train-test gap (i.e., exposure bias), rather than a separate, formal theoretical framework.**
>
> We provided the following analyses in the paper, which are sufficient to support the claim made in the introduction:
>
> -  **Remark on Mitigating Exposure Bias (Lines 265-298):** We point out that TPA mitigates exposure bias by expanding the alignment space from $O(|V|)$ to $O(|V|^2)$. This larger space significantly increases the probability of sampling paths that align with the student's own generation trajectory, thereby reducing the discrepancy between the teacher-forced training and the student's autoregressive inference. A more detailed discussion and analysis of this point are provided in Appendix D.1 (Lines 1108-1118).
>
> -  **More Discussion (Lines 1155-1190):** We analyze the benefits of TPA from the perspective of an ideal objective for sequence-level distillation. This discussion provides additional conceptual support for why TPA is an effective technique.
>
> ---
>
> > **Q2: Ignoring discussion on unified multimodal model.**
>
> We thank the reviewer for their valuable feedback. **The MLLMs we focus on are representative and widely adopted architectures; in fact, almost all prior MLLM knowledge distillation works have focused on this type of architecture, such as LLaVA-MoD [1], LaVA-KD [2], Align-KD [3] and MoVE-KD [4].**
>
> We understand the reviewer’s interest in exploring the boundaries of our method’s applicability. Since the term “projector-based multimodal language models” may be ambiguous, we provide the following discussion:
>
> **(1) If "projector" refers to the vision-language projector used for aligning vision and language modalities:** Our method operates on output token-level alignment and is thus agnostic to the low-level architectural details. This means our approach is applicable whether or not a model incorporates a vision-language projector.
>
> **(2) If "projector" refers to the output head that projects continuous embeddings to the discrete vocabulary:** Our method is currently designed for architectures that operate in a discrete output space. Unified multimodal models involve predictions in a continuous space, and our framework is not yet equipped to handle distillation for such models. **Indeed, most prior works [1-4] have also focused primarily on discrete output space.** We recognize this as a valuable direction for future research and have added a discussion on this point in our revised manuscript (Lines 534-536). We are grateful to the reviewer for bringing this to our attention.
>
> [1] Shu, Fangxun, et al. "Llava-mod: Making llava tiny via moe knowledge distillation." ICLR, 2025
>
> [2] Cai, Yuxuan, et al. "Llava-kd: A framework of distilling multimodal large language models." ICCV, 2025.
>
> [3] Feng, Qianhan, et al. "Align-KD: Distilling Cross-Modal Alignment Knowledge for Mobile Vision-Language Large Model Enhancement." CVPR, 2025.
>
> [4] Cao, Jiajun, et al. "Move-kd: Knowledge distillation for vlms with mixture of visual encoders." CVPR. 2025.

---

> ### Author Response · Authors · 2025-11-21
> **Response to Reviewer fYcx (2/4)**
>
> > **Q3: Code release.**
>
> We thank the reviewer for the comment. We have added a reproducibility statement in the revised manuscript (Lines 540-557). We promise to release our code and model weights. The repository link will be included in the final camera-ready version.
>
> ---
>
> > **Q4: Differences between Equations 2 and 5 on the Vanilla KD loss.**
>
> We thank the reviewer for this comment. We would like to clarify that the descriptions of the vanilla KD loss in Equations 2 and 5 are consistent. The apparent difference is that we omitted the prefix condition in Equation 5, **which has been explained in the text directly following Equation 5.** This was done to improve clarity, since Equation 5 is too long. By omitting the common prefix, we intended to make the comparison between the two main terms in Equation 5 more intuitive.
>
> ---
>
>
> > **Q5: The performance of the teacher models should be transferred to the main text.**
>
>
> We thank the reviewer for this constructive suggestion. Initially, we moved the table reporting the teacher's performance to the appendix due to space constraints. **We have now relocated it to the "Implementation Details" section in the main text as requested. (Table 2, Lines 337-341)**
>
> Furthermore, to provide a more intuitive comparison of the performance gap between the teacher and the distilled student models, we have added the following summary table to the Appendix E.3 (Lines 1239-1246). This table directly illustrates the performance drop after distillation. We sincerely thank the reviewers' comments, which helps improve the quality of our paper.
>
> | Type    | LLM        | GQA  | SQA  | TextVQA | POPE | MME  | MMB  | AVG  |
> | ------- | ---------- | ---- | ---- | :-------: | ---- | ---- | ---- | ---- |
> | Teacher | Qwen2-7B   | 64.6 | 80.7 | 64.1    | 86.1 | 78.6 | 76.0 | 75.1 |
> | Student | Qwen2-1.5B | 62.9 | 71.4 | 65.1    | 86.1 | 75.6 | 71.8 | 72.2 |
> | Student | Qwen2-0.5B | 60.4 | 60.7 | 59.9    | 86.8 | 68.9 | 63.2 | 66.7 |
> | Teacher | Qwen3-8B   | 64.0 | 83.4 | 64.1    | 86.4 | 80.4 | 77.3 | 76.0 |
> | Student | Qwen3-1.7B | 62.6 | 76.5 | 67.1    | 86.6 | 73.4 | 75.2 | 73.6 |
> | Student | Qwen3-0.6B | 61.2 | 68.4 | 64.1    | 86.9 | 70.0 | 67.6 | 69.7 |
>
> ---
>
>
> > **Q6: The efficiency analysis seems redundant.**
>
> We thank the reviewer for this comment. As distillation is inherently aimed at training smaller models, in Table 5 (revised manuscript), we compare the inference efficiency and performance of our distilled model against a larger 7B model to illustrate the advantages of using a compact model. This type of comparison is also adopted in prior distillation studies, such as LLaVA-MoD [1]. In addition, Table 6 (revised manuscript) presents the training efficiency, which, combined with inference results in Table 5, forms our overall efficiency analysis.
>
> [1] Shu, Fangxun, et al. "Llava-mod: Making llava tiny via moe knowledge distillation." ICLR, 2025

---

> ### Author Response · Authors · 2025-11-21
> **Response to Reviewer fYcx (3/4)**
>
> > **Q7: Compared with $y _ k$ and $y _ {k+1}$ and are both from the ground-truth response.**
>
> We appreciate the reviewer's comment. However, when $y _ k$ and $y _ {k+1}$ are both sampled from the ground-truth response, the proposed alignment objective simplifies to the objective of Vanilla KD, which has been dicussed in our paper already.
>
> ---
>
>
> > **Q8: Sample from the teacher model instead of student model to compute the transition probability.**
>
> We thank the reviewer for this insightful question. As we discussed in Lines 261-264, we chose to sample from the student model instead of the teacher model for the following primary reasons:
>
> 1. **On-policy Exploration:** Sampling from the student distribution allows for on-policy exploration of the student’s own predictive space. This is crucial for identifying and correcting the student's potential errors. Conversely, the teacher model, due to its high quality, produces samples that largely overlap with the ground truth, leading to inefficient exploration of the areas where the student model actually struggles. This strategy of on-policy sampling from the model being trained has been proven effective in prior works [1-2].
>
> 2. **Computational Efficiency:** While sampling from either the student or the teacher model requires an additional forward pass, sampling from the student is more computationally efficient. As shown in the table below, using the teacher model for sampling increases the training time by approximately 1.3x compared to sampling from the student.
>
>    | Sampled Source | Training Time (H) |
>    | :------------: | :-------------------: |
>    |    Student     |          509          |
>    |    Teacher     |          682          |
>
> [1] Agarwal, Rishabh, et al. "On-policy distillation of language models: Learning from self-generated mistakes." ICLR. 2024.
>
> [2] Lu, Kevin, et al. "On-Policy Distillation." Thinking Machines Lab: Connectionism, 2025.
>
> ---
>
>
> > **Q9: Cancel the vanilla KD loss.**
>
> We thank the reviewer for this insightful suggestion. Following the reviewer's request, we have conducted an experiment to evaluate the model's performance without the vanilla KD loss, relying solely on our proposed $\mathcal{L} _ \mathrm{iva}$ and $\mathcal{L} _ \mathrm{tpa}$ alignment losses. We have included this result in Table 15 of our revised manuscript, alongside the ablation studies on the impact of different loss components requested by Reviewer SRQm.
>
> As shown in the table below, we observe a slight performance drop when the vanilla KD loss is removed (the average score decreases from 66.7 to 66.6). This result indicates that our proposed IVA and TPA can effectively mitigate the performance degradation caused by removing the vanilla KD loss, demonstrating strong alignment capabilities.
>
> |                | GQA      | SQA  | TextVQA | POPE     | MME      | MMB      | AVG      |
> | -------------- | -------- | ---- | :-------: | -------- | -------- | -------- | -------- |
> | Align-TI       | **60.4** | 60.7 | 59.9    | **86.8** | **68.9** | **63.2** | **66.7** |
> | w/o Vanilla KD | 60.3     | 60.4 | 60.6    | 86.4     | 68.5     | 63.1     | 66.6     |
>
> Moreover, in our framework, vanilla KD can be viewed as aligning the initial state (next-token distribution), while TPA further aligns transition probabilities on top of this. We regard the initial-state alignment and transition-probability alignment as a unified mechanism within our approach.

---

> > ### Author Response · Authors · 2025-11-21
> > **Response to Reviewer fYcx (4/4)**
> >
> > > **Q10: Questions need to be clarified 1,2,4. (About performance reported in Table 2 and 3.)**
> >
> > We thank the reviewer for the careful review of our paper. We would like to clarify a potential misunderstanding regarding our experimental setup, which appears to be the root of the three questions raised. It appears that the reviewer may have misinterpreted our reference to 'Comparison with Distillation Strategy **Designed for LLMs**'(Line 355, 366) as the distillation experiments were **conducted on LLMs**.
> >
> > (1)  The experiments in Table 2 were not conducted on text-only LLMs. Instead, Table 2 compares our method with several distillation strategies that were designed for LLMs. We included this to make our paper more comprehensive. Since MLLMs are built upon an LLM foundation, these strategies can be readily adapted to MLLMs. Additionally, we do not implement IVA on LLMs.
> >
> > (2)  Regarding the performance consistency across Tables 1, 2, and 3: The reviewer's intuition is correct. The performance of our full method, Align-TI (w/ IVA and TPA), should be the same across these tables, and it is. The confusion stems from the previous misunderstanding that Table 2 reported results on LLMs.
> >
> > (3)  Regarding the performance in Tables 6, 7, and 8: These tables present a detailed ablation study specifically for the IVA component. It can be seen that the performance reported in Tables 6, 7, and 8 matches precisely the result for the "w/ IVA only" configuration presented in the ablation study in Table 3.
> >
> > **Notes:** The table numbers here are from before the revision.
> >
> > ---
> >
> >
> > > **Q11: Layer 0 also achieves a good performance.**
> >
> > We thank the reviewer for the insightful observation. However, this result does not contradict our claim.
> >
> > As shown in Table 6, the highest IRS score occurs at layer 21, which achieves the best performance overall. While layer 0 also shows relatively good performance, it is still lower than that of the higher-IRS layers such as 14 and 21. We also investigated why layer 0 performs reasonably well. As visualized in Appendix Fig. 9, we found that layer 0’s attention predominantly focuses on regions with large visual token index values. Due to the modeling characteristics of LLM’s causal attention, these tokens, although not instruction-related, can still play an important role. This observation is consistent with findings reported in [1].
> >
> > Why IRS important: Our proposed IRS provides a quantitative measure of the correlation between instructions and the attention maps, thereby guiding IVA to align using the most instruction-relevant Attention maps. The motivation for introducing IRS is to avoid experience-based layer selection. Additionally, this approach was endorsed by Reviewer vd3n, who noted, *"The IRS metric is a good method for selecting the most relevant layer, rather than relying on manual design."*
> >
> > [1] Wen, Zichen, et al. *Token Pruning in Multimodal Large Language Models: Are We Solving the Right Problem?* ACL (2025).
> >
> > ---
> >
> > We would be more than happy to discuss any further questions!

---

> > > ### Comment · Reviewer_fYcx · 2025-11-25
> > > **Response**
> > >
> > > Thank you for your detailed response addressing my concerns. I also have the following questions and suggestions. First, I think you should explicitly include the condition $x, y_{<k}$. This term does not significantly lengthen the equation but provides a clearer comparison between vanilla KD and the transition term. They have different conditions. Second, could you explain why, in TextVQA, the student model achieves higher performance than the teacher model? For efficiency, could you explicitly highlight the time cost of the parallelized calculation? This might be a key factor in making your method practical, particularly in terms of time savings. Regarding Q7 and Q8, could you analyze the performance results? And can you explain why a high-quality $y_k$ does not lead to higher performance than the student’s on-policy exploration? Does the current method highlight the student model’s prediction error? Consequently, the results for Q9 are also interesting. When next-token distillation (vanilla KD) is removed, the student model is still significantly boosted through the transition term, showing only a very slight performance drop. However, the transition is highly conditioned on the next-token prediction of the student model. Could you explain this in depth? I am willing to raise my score if you could address the concerns above.

---

> > > > ### Author Response · Authors · 2025-11-27
> > > > **Response to Reviewer fYcx (1/3)**
> > > >
> > > > We sincerely thank the reviewer again for reading our rebuttal and for the time and effort dedicated to reviewing our work. We are also deeply grateful for your willingness to reconsider the rating. We appreciate the insightful questions and suggestions, which is helpful in improving our paper.
> > > >
> > > > ---
> > > >
> > > > >  **Q1: Explicitly include the condition $\boldsymbol{x}, \boldsymbol{y} _ {1:L}^{\mathcal{D}}$**
> > > >
> > > > We thank the reviewer for this insightful suggestion. Initially, we omitted the common condition $\boldsymbol{x}, \boldsymbol{y}_{1:L}^{\mathcal{D}}$ to highlight the distinction between Initial State Alignment and Transition Probability Alignment, and to maintain the conciseness of the formula. However, we agree that this omission could potentially lead to misunderstanding. **Therefore, we have revised Equation 5 to explicitly include this condition.** We have also verified the consistency of this notation throughout the manuscript, including updates to Equation 7 and the algorithm description in Appendix B.6. We thank you again for helping us improve the clarity of our paper.
> > > >
> > > > ---
> > > >
> > > > >  **Q2: Why the student model achieves higher performance than the teacher model on TextVQA?**
> > > >
> > > > Thank you for this insightful observation. We find that this phenomenon, where the student model outperforms the teacher, is specific to the TextVQA benchmark. On benchmarks like GQA, SQA, and MME, the teacher model consistently achieves higher performance. This leads us to believe that **the student's superior performance is linked to the inherent characteristics of the TextVQA benchmark.**
> > > >
> > > > **First, we analyzed the nature of the TextVQA benchmark.** It is a visual question-answering task focused on text within images. A critical feature of its evaluation protocol is the inclusion of a reference field, which contains all the OCR tokens extracted from the image. This feature substantially lowers the difficulty of the task. As illustrated by the example below, a model can often answer correctly simply by locating the relevant region mentioned in the question and then finding the corresponding text in the reference field.
> > > >
> > > > ```
> > > > What is the number on the runner in middle?
> > > > Reference OCR token: money, 57859, adida, 6531, adidas, mone
> > > > Answer the question using a single word or phrase.
> > > > ```
> > > >
> > > > Given this, we hypothesize the primary reasons is: **the 7B teacher model is over-parameterized for the TextVQA benchmark.**  Since the task's complexity is reduced, a model with such a large capacity is not necessary to achieve high performance. Therefore, with the help of distillation-based training, the student model was trained better, and the 2B student model also performed very well. **A similar outcome was reported in the LLaVA-MoD[1], where their student model also surpassed the teacher model's performance on TextVQA.**
> > > >
> > > > **We have added this analysis to Appendix E.3 (Lines 1258-1261) in our revised manuscript to provide further clarification.** Thank you again for pointing this out.
> > > >
> > > > [1] Shu, Fangxun, et al. "Llava-mod: Making llava tiny via moe knowledge distillation." ICLR, 2025

---

> ### Author Response · Authors · 2025-11-27
> **Response to Reviewer fYcx (2/3)**
>
> > **Q3: Efficiency of parallelized calculation.**
>
> We thank the reviewer for this insightful comment regarding the efficiency of our method. We have provided the empirical time cost of TPA using parallelized calculation in Table 6 of our paper, and we summarize the result of TPA below:
>
> | Method                            | Training Time (H)  | Memory (GiB) |
> | --------------------------------- | ------------------ | ------------ |
> | Vanilla KD                        | 355                | 70.6         |
> | TPA (w/ parallelized calculation) | 504 ($\times 1.4$) | 75.3         |
>
> Furthermore, we understand the reviewer is interested in the time savings achieved through parallelization, which requires a comparison of TPA's training cost with and without this technique. Accurately measuring the empirical training time for TPA without parallelization is difficult. Therefore, **we offer a theoretical comparison based on the number of required forward passes, which is a primary driver of the overall time cost.** Considering a training dataset with $N$ samples, an average response length of $L$, and the number of sampled tokens $d$, the comparison of forward pass overhead is as follows:
>
> | Method                    | Student Model Forward Passes | Teacher Model Forward Passes |
> | :------------------------ | :--------------------------: | :--------------------------: |
> | Vanilla KD                |             $N$              |             $N$              |
> | TPA (w/ Parallelization)  |             $2N$             |             $N$              |
> | TPA (w/o Parallelization) |          $dLN + N$           |            $dLN$             |
>
> By introducing the ribbon attention mask, TPA achieves parallelized calculation, ensuring that each token attends to its correct prefix. This allows us to obtain probability estimates for all tokens within a sample in a single forward pass. In contrast, without parallelization, estimating the probability for each token would require a distinct path and a separate forward pass, leading to $dLN + N$ forward pass for the student, which is computationally impractical.
>
> **We have added this detailed analysis to Appendix D.2 to clarify the significant efficiency gains and practicality of our method.** We thank the reviewer for this constructive comment, which has helped improve the clarity of our paper.

---

> > ### Author Response · Authors · 2025-11-27
> > **Response to Reviewer fYcx (3/3)**
> >
> > > **Q4: Student’s on-policy exploration v.s. teacher's high-quality**
> >
> > We thank the reviewer for this insightful comment. We understand the reviewer's concern regarding why the student's low-quality on-policy exploration yields better results than the teacher's high-quality data. We would like to address this from two perspectives:
> >
> > **(1) Standard Practice in KD:** It is a widely accepted consensus in the KD community to sample from the student rather than the teacher during the training phase. Since the teacher model is typically frozen, it is standard practice to use it for off-policy data processing (often offline) to maximize efficiency. Online sampling from the teacher during training is computationally expensive and rarely adopted in existing literature.
> >
> > **(2) Advantages of On-Policy Exploration:** Sampling from the student for on-policy exploration during training offers several advantages, which have been validated by increasingly more recent works [1-3]:
> >
> > *   **Learning from self-generated mistakes:** Under the on-policy framework, the student model is permitted to explore its own output distribution. This inevitably leads to the generation of sub-optimal or even erroneous tokens. **When these sequence prefixes containing "mistakes" are presented to the teacher model, the teacher acts as a corrector.** Specifically, if the student tends to generate an incorrect next token at a specific generation step, the teacher—when exposed to this context—will guide the distribution to avoid sustaining the error. This corrective knowledge is then transferred to the student via distillation. This mechanism is discussed in GKD [1].
> >
> > *   **Mitigating Exposure Bias:** Furthermore, performing on-policy sampling on the student's distribution exposes the model to its own test-time distribution during the training phase. This will help bridge the gap between training and test distributions, leading to better generalization.
> >
> > The current method does not explicitly design a mechanism to "highlight" the student model’s prediction error. However, the fundamental premise of KD is that the student's prediction error is inherently larger than the teacher's. Therefore, the distillation process naturally leverages this discrepancy to transfer superior knowledge from the teacher to the student.
> >
> > [1] Agarwal, Rishabh, et al. "On-policy distillation of language models: Learning from self-generated mistakes." ICLR. 2024.
> >
> > [2] Lu, Kevin, et al. "On-Policy Distillation." Thinking Machines Lab: Connectionism, 2025.
> >
> > [3] Ye, Tianzhu, et al. "Black-Box On-Policy Distillation of Large Language Models." arXiv:2511.10643 (2025).
> >
> > ---
> >
> > >  **Q5: When Vanilla KD is removed, the student model is still significantly boosted through the transition term.**
> >
> > We thank the reviewer for this insightful observation. You are correct that TPA relies on a reasonable next-token prediction distribution from the student model, as TPA involves sampling from the student's distribution. If the student fails to provide a reasonable initial distribution, TPA would indeed be ineffective. **However, it is important to note that our final training objective includes the SFT Loss alongside IVA, TPA, and Vanilla KD.** The SFT loss directly aligns the student's next-token prediction with the Ground Truth, thereby guaranteeing the fundamental quality of the next-token prediction required for TPA to function effectively.
> >
> > Furthermore, **TPA implicitly captures some of the alignment objectives of Vanilla KD due to the sequential nature of the process.** At step $k$, Vanilla KD aligns $y_k$ conditioned on $x, y_{<k}^{\mathcal{D}}$. TPA aligns the transition $y_{k+1} | y_k$ conditioned on $x, y_{<k}^{\mathcal{D}}$. Crucially, at the previous step $k-1$, TPA aligns $y_k | y_{k-1}$ conditioned on $x, y_{<k-1}^{\mathcal{D}}$. Since $y_{k-1}$ is sampled from the student, there is a probability that the sampled token matches the ground truth token $y_{k-1}^{\mathcal{D}}$. When this occurs, the optimization direction aligns with that of Vanilla KD. However, since sampling the ground truth token $y_{k-1}^{\mathcal{D}}$ is not guaranteed via the student's distribution, this implicit alignment is not strictly equivalent to explicit Vanilla KD. This explains why explicitly including the Vanilla KD term still results in performance improvements on specific benchmarks (e.g., +0.3 on SQA and +0.4 on MME), as observed in our results.
> >
> > ---
> >
> > Thank you again for your valuable contribution to improving our work. We are happy to discuss any further concerns.

---

### Official Review · Reviewer_Kx6f · 2025-10-30

**Soundness:** 3
**Presentation:** 3
**Contribution:** 3
**Rating:** 6
**Confidence:** 4

**Summary:**

This paper introduces a new knowledge distillation framework for multimodal large language models (MLLMs). Traditional distillation methods align student and teacher outputs at the next-token level, but they overlook dynamic token interactions essential for multimodal understanding and coherent generation.
Align-TI models distillation from the perspective of token interactions and introduces two components, i.e., Instruction-aware Vision Alignment (IVA) and Transition Probability Alignment (TPA). Experiments across standard benchmarks show that Align-TI significantly outperforms both vanilla KD and larger MLLMs (e.g., LLaVA-1.5-7B) while offering better computational efficiency. Ablation and scaling studies confirm that IVA and TPA contribute complementary benefits, making Align-TI a robust and efficient approach for distilling high-performing small-scale MLLMs.

**Strengths:**

1. The shift from static next-token alignment to token interaction modeling (IVA and TPA) provides a theoretically grounded and empirically validated innovation in knowledge distillation.
2. Extensive experiments and ablations across multiple benchmarks demonstrate robustness and scalability, including efficiency analysis and architectural generalization.
3. The proposed TPA component explicitly addresses train-test distribution discrepancies, a critical and underexplored issue in multimodal distillation.

**Weaknesses:**

1. Essentially, MLLMs already perform rich cross-modal and intra-modal token interactions through their attention layers. From this perspective, Align-TI appears to be an incremental refinement of existing attention mechanisms, i.e., reweighting visual focus (IVA) and regularizing output dynamics (TPA). It would be valuable to see further discussion or empirical evidence clarifying how Align-TI captures interactions beyond what standard attention already provides.
2. The evaluation primarily uses image-text benchmarks. It remains unclear how Align-TI generalizes to other modalities or other multimodal tasks, such as the video-based understanding benchmark, which would strengthen claims of broad multimodal applicability.

**Questions:**

Please refer to the Weaknesses.

---

> ### Author Response · Authors · 2025-11-21
> **Response to Reviewer Kx6f**
>
> We sincerely appreciate your constructive comments and valuable suggestions. Thank you for the time and effort you have devoted to assessing our manuscript and helping us improve its quality. Our detailed responses to your concerns are provided below.
>
> ---
>
>  > **Q1: It would be valuable to see further discussion or empirical evidence clarifying how Align-TI captures interactions beyond what standard attention already provides.**
>
> We thank the reviewer for their insightful comments. We agree that MLLMs already perform rich cross-modal and intra-modal token interactions through their attention layers. However, we would like to clarify that our proposed Align-TI is a **distillation method**. Its goal is to enable a student model to learn from a teacher model that exhibits more precise and effective token interactions.
>
> To directly address the reviewer's question about *how Align-TI captures interactions beyond what the student's standard attention already provides*, **we have added a new section, Section 4.3: Analysis on IVA and TPA, to the paper.** This section provides the following empirical evidence:
>
> **(1) IVA:** We have included a new visualization analysis of the Vision-Instruction Attention Map (**Figure 8, Lines 455-475**) to compare the student model's attention before and after applying IVA distillation. We observe that after distillation, the student’s attention maps become significantly closer to the teacher’s attention maps. Specifically, we identify two primary improvements:
>
> *   **Focus Correction:** Without IVA, the student may incorrectly attend to unrelated objects. For instance, when asked about a "green logo," it focuses on an entirely different logo (top row). IVA helps redirect its attention to the correct target, mirroring the teacher.
> *   **Focus Sharpening:** Even when the student localizes the correct general area without IVA, their attention can be dispersed across irrelevant regions. IVA refines this into a concentrated map that closely follows the teacher’s precise focus (bottom row).
>
> These findings demonstrate that IVA effectively distills the teacher's superior ability to ground instructions in visual information, capturing a more accurate interaction pattern than the student model achieved on its own.
>
> **(2) TPA:** As discussed in our manuscript (**Figure 7, Lines 477-485**), we use the `%ExAccErr` metric to measure the mitigation of exposure bias. The results show that TPA significantly reduces this error. This indicates that the student successfully learns the teacher's more stable output dynamics, effectively capturing the nuanced token-to-token dependencies within a response that the teacher model has mastered.
>
> We are grateful for the reviewer's valuable feedback, which has helped us substantially improve the clarity and quality of our paper.
>
> ---
>
> > **Q2: Remains unclear how Align-TI generalizes to other modalities or other multimodal tasks.**
>
> We thank the reviewer for their valuable feedback. We did not evaluate on other multimodal tasks, such as video-based understanding benchmarks, because our 3.6M training dataset consists exclusively of image-text data. Consequently, our evaluations were focused on image-text benchmarks. **We would like to note that this evaluation scope is consistent with several prior MLLM distillation works that also concentrated on image-text benchmarks, including LLaVA-MoD [1], LLaVA-KD [2], Align-KD [3] and MoVE-KD [4].**
>
> However, we agree with the reviewer that evaluating on a wider range of multimodal tasks is necessary to strengthen the claims of broad multimodal applicability. **Therefore, we have added this point to the Limitations and Future Work section of our paper (Lines 532-534).** We sincerely appreciate the reviewers bringing this limitation to our attention.
>
> [1] Shu, Fangxun, et al. "Llava-mod: Making llava tiny via moe knowledge distillation." ICLR, 2025
>
> [2] Cai, Yuxuan, et al. "Llava-kd: A framework of distilling multimodal large language models." ICCV, 2025.
>
> [3] Feng, Qianhan, et al. "Align-KD: Distilling Cross-Modal Alignment Knowledge for Mobile Vision-Language Large Model Enhancement." CVPR, 2025.
>
> [4] Cao, Jiajun, et al. "Move-kd: Knowledge distillation for vlms with mixture of visual encoders." CVPR. 2025.
>
> ---
>
> We would be more than happy to discuss any further questions!

---

### Official Review · Reviewer_SRQm · 2025-10-31

**Soundness:** 3
**Presentation:** 3
**Contribution:** 3
**Rating:** 6
**Confidence:** 5

**Summary:**

This paper presents Align-TI, a knowledge distillation framework for MLLMs. By introducing Instruction-aware Vision Alignment (IVA) and Transition Probability Alignment (TPA), it aligns a student model's ability to extract instruction-relevant visual information and dynamic generation logic with a teacher's. The proposed method achieves state-of-the-art results across multiple benchmarks, proving effective for enhancing smaller MLLMs.

**Strengths:**

1.Extensive experiments demonstrate state-of-the-art performance, while thorough ablation studies validate the method's generalization across different MLLM architectures.
2.The insight is interesting. The work provides an interesting perspective by addressing the often-overlooked problem of insufficient visual token distillation in existing MLLM knowledge distillation methods.
3.The proposed IVA and TPA components are novel and effective, successfully aligning the student’s visual attention and generation logic with the teacher’s.
4.The paper is well-organized and easy to follow.

**Weaknesses:**

1.A potential direction for future work could be applying this distillation approach during the pre-training stage. It would be insightful to discuss how this might enhance the vision-language alignment.
2.The paper could be further strengthened by a more detailed analysis of the individual loss components. For instance, an ablation study quantifying the specific impact of each term ($L_{sft}, L_{kd}, L_{iva}, L_{tpa}$) would offer a clearer understanding of their respective contributions to the final performance.

**Questions:**

See Weakness

---

> ### Author Response · Authors · 2025-11-21
> **Response to Reviewer SRQm**
>
> We sincerely appreciate your constructive comments and valuable suggestions. Thank you for the time and effort you have devoted to assessing our manuscript and helping us improve its quality. Our detailed responses to your concerns are provided below.
>
> ---
>
> > **Q1: Dicussion on Align-TI's potential of vision-language alignment during the pre-training stage.**
>
> We thank the reviewer for the insightful suggestion. In this paper, we do not introduce a KD objective during pre-training, relying solely on the SFT objective. This design was primarily to ensure a fair comparison with the baseline method (LLaVA-MoD [1]).
>
> However, we fully agree that the direction pointed out by the reviewer holds great potential. In fact, recent work such as LLaVA-KD [2] has already demonstrated that introducing a KD objective during the pre-training stage to train the adaptor can enhance vision-language alignment. Furthermore, as reviewer dQg7 mentioned, Align-KD [3] also focuses on the alignment of the adaptor. This inspires us that strengthening the training of the adaptor could be key to further improving the model's understanding of visual concepts. **We have included this as a direction for future work in our "Limitations and Future Work" section (Line 536-537).** We sincerely appreciate the reviewer bringing this to our attention.
>
> [1] Shu, Fangxun, et al. "Llava-mod: Making llava tiny via moe knowledge distillation." ICLR, 2025
>
> [2] Cai, Yuxuan, et al. "Llava-kd: A framework of distilling multimodal large language models." ICCV, 2025.
>
> [3] Feng, Qianhan, et al. "Align-KD: Distilling Cross-Modal Alignment Knowledge for Mobile Vision-Language Large Model Enhancement." CVPR, 2025.
>
> ---
>
>
> > **Q2: Detailed analysis of the individual loss components.**
>
> We thank the reviewer for this insightful suggestion. **To clarify the contribution of each loss component, we have conducted a detailed ablation study as suggested by the reviewer.** The results and corresponding analysis are now included in a new subsection, "Loss Contribution Analysis," in the revised manuscript (Lines 1206-1226).
>
> The study, presented in the new table, systematically evaluates the impact of each loss term: Supervised Fine-Tuning ($\mathcal{L} _ {\mathrm{sft}}$), Instruction-aware Vision Alignment ($\mathcal{L} _ {\mathrm{iva}}$), Vanilla KD ($\mathcal{L} _ {\mathrm{kd}}$), and Transition Probability Alignment ($\mathcal{L} _ {\mathrm{tpa}}$). Our key findings are as follows:
>
> - Starting from the baseline with only $\mathcal{L} _ {\mathrm{sft}}$, adding $\mathcal{L} _ {\mathrm{iva}}$ or $\mathcal{L} _ {\mathrm{kd}}$ individually yields average improvements of 0.8 and 0.7, respectively, while combining the two results in a larger gain of 1.3.
>
> - When initiating from a Vanilla KD configuration, incorporating $\mathcal{L} _ {\mathrm{iva}}$ and $\mathcal{L} _ {\mathrm{tpa}}$ enhances performance by 0.6 and 1.4, respectively. Applying both losses together achieves a total enhancement of 1.7.
>
> - The absence of $\mathcal{L} _ {\mathrm{kd}}$ in this setup only slightly decreases performance by 0.1, suggesting that the other components can effectively compensate for its omission.
>
> This new ablation study provides a quantitative and clear understanding of how each component contributes to the final performance, thereby strengthening the paper as the reviewer suggested.
>
>
> | Loss                                                         | GQA      | SQA      | TextVQA  | POPE     | MME      | MMB      | AVG      |
> | ------------------------------------------------------------ | -------- | -------- | :--------: | -------- | -------- | -------- | -------- |
> | $\mathcal{L} _ {\mathrm{sft}}$                               | 57.6     | 59.0     | 59.1     | 86.2     | 66.1     | 57.9     | 64.3     |
> | $\mathcal{L} _ {\mathrm{sft}} + \mathcal{L} _ \mathrm{iva}$  | 59.6     | 58.0     | **61.3** | 86.5     | 66.8     | 58.3     | 65.1     |
> | $\mathcal{L} _ {\mathrm{sft}} + \mathcal{L} _ \mathrm{kd}$   | 59.3     | 59.7     | 59.2     | 86.2     | 65.0     | 60.4     | 65.0     |
> | $\mathcal{L} _ {\mathrm{sft}} + \mathcal{L} _ \mathrm{iva} + \mathcal{L} _ \mathrm{kd}$ | 59.8     | 59.4     | 60.5     | 86.5     | 66.3     | 61.3     | 65.6     |
> | $\mathcal{L} _ {\mathrm{sft}} + \mathcal{L} _ \mathrm{kd} + \mathcal{L} _ \mathrm{tpa}$ | 60.3     | **61.0** | 59.6     | 86.5     | 68.1     | 63.0     | 66.4     |
> | $\mathcal{L} _ {\mathrm{sft}} + \mathcal{L} _ \mathrm{tpa} + \mathcal{L} _ \mathrm{iva}$ | 60.3     | 60.4     | 60.6     | 86.4     | 68.5     | 63.1     | 66.6     |
> | $\mathcal{L} _ {\mathrm{sft}} + \mathcal{L} _ \mathrm{iva} + \mathcal{L} _ \mathrm{kd} + \mathcal{L} _ \mathrm{tpa}$ | **60.4** | 60.7     | 59.9     | **86.8** | **68.9** | **63.2** | **66.7** |
>
> ---
>
> We would be more than happy to discuss any further questions!

---

> > ### Comment · Reviewer_SRQm · 2025-11-25
> >
> > Thanks for the response. The rebuttal clearly resolves my concern about how each loss component contributes. Considering the method's insight, novelty, and the solid experimental results, I am convinced of its value to the community. Consequently, I will raise my score from 6 to 8 and recommend accepting this paper.

---

> > > ### Author Response · Authors · 2025-11-25
> > > **Official Comment by Authors**
> > >
> > > Dear Reviewer SRQm,
> > >
> > > We sincerely thank you for your recognition of our work and for raising the score. Your positive feedback is greatly encouraging to us. We also deeply appreciate the time and effort you dedicated to reviewing our manuscript, and your insightful comments are helpful to us.
> > >
> > > Sincerely, Paper #9359 Authors

---

### Official Review · Reviewer_vd3n · 2025-11-01

**Soundness:** 3
**Presentation:** 4
**Contribution:** 2
**Rating:** 4
**Confidence:** 4

**Summary:**

This paper proposes Align-TI, a new knowledge distillation (KD) framework for Multimodal Large Language Models (MLLMs). The framework is composed of two new components: (1) Instruction-aware Vision Alignment (IVA) and (2) Transition Probability Alignment (TPA).

**Strengths:**

The insight for IVA is strong. Recognizing that many visual tokens are redundant and that distillation should focus on instruction-salient regions is a valuable contribution.

The paper provides a principled exploration of different transformer layers for visual-text attention distillation. The IRS metric is a good method for selecting the most relevant layer, rather than relying on manual design.

The TPA component's objective is practical. Aligning the full vocabulary's transition matrix is infeasible, so the insight to focus only on a sampled, high-probability set of tokens is a sensible and efficient approximation.

**Weaknesses:**

- The two proposed modules, IVA and TPA, feel separate and not well-integrated. IVA is a VLM-specific technique for aligning visual-instruction interactions. TPA, however, is a general-purpose LLM distillation method for text generation. The paper does not convincingly unify them into a single coherent KD framework.

- The paper is difficult for a reader unfamiliar with the field to assess. The related work section is in the appendix, so the main text lacks a necessary discussion of previous work. Without this, it is hard to judge the novelty of IVA against other attention distillation methods or TPA against other sampling-based KD strategies.

- The objective for TPA is unclear. The notation y(k+1) conditions on y(k) misleadingly suggests a 2-gram (bigram) model, which seems to contradict the complex attention mechanisms of an LLM. This makes the core objective difficult to understand.

- There appears to be a misalignment between the goal of TPA (aligning transition matrices) and its implementation (a sampling algorithm). This ambiguity makes it hard to justify the novelty. The authors should clarify if the sampling strategy itself is new for LLM KD.

- The performance claims are not fully supported. Ablation: The performance gain from IVA is not significant. Table 3 shows adding IVA on top of TPA only provides a minimal 0.3-point gain (66.4 to 66.7), suggesting TPA does all the heavy lifting. Efficiency: TPA adds significant training overhead (509 / 355, about 1.43x). A critical baseline is missing: training Vanilla KD for the same amount of time as TPA (e.g., on 1.43x more data). It is possible that Vanilla KD could match the performance of TPA if given the same computational budget.

The paper proposes two methods that seem disjointed: one for VLM attention and one for general LLM generation. The novelty of the TPA component is difficult to justify due to a confusing formulation and a misalignment between its objective and its implementation. Finally, the empirical gains from the main IVA component are not significant, and the overall framework lacks a fair comparison against an efficiency-equivalent baseline.

**Questions:**

NA. Please refer to the weaknesses.

---

> ### Author Response · Authors · 2025-11-21
> **Response to Reviewer vd3n (1/3)**
>
> We sincerely appreciate your constructive comments and valuable suggestions. Thank you for the time and effort you have devoted to assessing our manuscript and helping us improve its quality. Our detailed responses to your concerns are provided below.
>
> ---
>
> > **Q1: IVA and TPA feel separate and not well-integrated.**
>
> We thank the reviewer for this insightful comment. We would like to clarify that IVA and TPA were intentionally designed as complementary modules within a unified KD framework, targeting the student-teacher gap at two distinct and critical stages of a MLLM's operation: the **Prefilling** stage and the **Decoding** stage. Specifically:
> **(1) IVA focuses on interactions during the Prefilling stage.** Its purpose is to distill the teacher model's sophisticated ability to process and fuse visual features in the context of a given instruction.
> **(2) TPA focuses on interactions during the Decoding stage.** It directly distills the generative behavior for the response tokens, making the student's output textually more consistent with the teacher's.
>
> Furthermore, **both IVA and TPA share the same ultimate goal**: to enable the student model to generate responses that are more closely aligned with those of the teacher model. IVA and TPA use different mechanisms to achieve this. TPA acts **directly** on the response tokens, while IVA acts **indirectly** by aligning the processing of visual tokens, which in turn facilitates better alignment of subsequent response tokens. **By addressing both the visual understanding (via IVA) and the textual generation (via TPA), our framework provides a more comprehensive distillation.** These materials are also described in the paper (Lines 205-209).
>
> Moreover, **we have added Sec. 3.3, Overall Objective for Align-TI (Lines 318–323)**, which shows that the two modules are jointly optimized by a combined objective function. We sincerely appreciate the reviewers' comments, which helps improve the quality of our paper.
>
> ---
>
> > **Q2: Related work is in the appendix.**
>
> We thank the reviewer for this comment. Due to the page limit, we moved this section to the Appendix.
>
> However, we have discussed several key related works and their limitations within the Introduction of the main text (Lines 80-92). This discussion covers recent methods such as LLaVADI[1], LLaVA-MoD[2], LLaVA-KD[3] and Align-KD[4]. By highlighting the limitations of these prior approaches, we clarify the distinctions and demonstrate the novelty of our proposed methods. This is sufficient to allow readers to judge the novelty of our work.
>
> Furthermore, to make the paper more accessible to readers unfamiliar with the field, we have included a dedicated Preliminaries section (Lines 126-157). This section provides the necessary background on foundational concepts, which helps readers unfamiliar with the field to better understand and assess our contributions.
>
> [1] Xu, Shilin, et al. "Llavadi: What matters for multimodal large language models distillation." arXiv preprint arXiv:2407.19409, 2024.
>
> [2] Shu, Fangxun, et al. "Llava-mod: Making llava tiny via moe knowledge distillation." ICLR, 2025
>
> [3] Cai, Yuxuan, et al. "Llava-kd: A framework of distilling multimodal large language models." ICCV, 2025.
>
> [4] Feng, Qianhan, et al. "Align-KD: Distilling Cross-Modal Alignment Knowledge for Mobile Vision-Language Large Model Enhancement." CVPR, 2025.

---

> ### Author Response · Authors · 2025-11-21
> **Response to Reviewer vd3n (2/3)**
>
> > **Q3: TPA suggests a 2-gram model.**
>
>
> We thank the reviewer for their valuable feedback. We believe there may be a misunderstanding regarding our proposed TPA objective, which we would like to clarify here.
>
> The reviewer’s interpretation of the notation $y _ {k+1}$ conditioned on $y _ k$ as representing a 2-gram (bigram) model stems from a simplification we made in the formula for readability. **As noted directly below Equation 5, we omitted the common prefix $\boldsymbol{x}$ and $\boldsymbol{y} _ {<k}^{\mathcal{D}}$ in the notation to better highlight the contrast between the Initial State Alignment $\mathcal{L} _ {\mathrm{kd}}(\theta)$ and the Transition Probability Alignment $\mathcal{L} _ {\mathrm{tpa}}(\theta)$.** We acknowledge that this simplification may have led to the misunderstanding. Therefore, we have revised Equation 5 to explicitly include this conditioning, making the formulation clearer.
>
> ---
>
> > **Q4: Misalignment between the goal of TPA and its implementation.**
>
> We thank the reviewer for this insightful comment.
>
> The TPA objective aims to align the token transition matrices between the teacher and student models. As the reviewer astutely pointed out in the Strengths section, *"Aligning the full vocabulary's transition matrix is infeasible, so the insight to focus only on a sampled, high-probability set of tokens is a sensible and efficient approximation."* Specifically, we use a Monte Carlo sampling approach to estimate the expectation of the TPA objective. **This is a standard and unbiased estimation strategy, widely used when direct computation of an expectation is intractable.**
>
> We would like to clarify that the combination of our proposed TPA objective and its estimation via a Monte Carlo sampling approach is indeed new for the field of MLLM KD. Moreover, the novelty of our proposed TPA is appreciated by reviewers Kx6f, dQg7 and SRQm.

---

> ### Author Response · Authors · 2025-11-21
> **Response to Reviewer vd3n (3/3)**
>
> > **Q5.1: The performance gain from IVA is not significant.**
>
> We thank the reviewer for this insightful question.
>
> First, we would like to clarify that IVA yields a more substantial improvement in other settings. Specifically, when applied on top of SFT, IVA achieves an average improvement of +0.8 across multiple benchmarks, which indicates that IVA is indeed effective as an independent component.
>
> Second, for the observed smaller gain when combining IVA with TPA, we have discussed this phenomenon in Lines 395–399. The reasons can be summarized as follows:
>
> - **Principle Perspective (Lines 395-397):** The core objective of MLLM KD is to align the student model's responses with the teacher model. TPA **directly** optimizes response-level alignment. In contrast, IVA operates **indirectly** by guiding the student to imitate the teacher's processing of visual tokens, thereby refining hidden representations for better response generation. When combined, **TPA's direct alignment leaves limited room for IVA's indirect optimization to yield substantial gains.**
>
> - **Overhead Perspective (Line 397-399):** While IVA's performance gain over TPA is limited, it can be integrated with TPA at **near-zero cost**. As shown below:
>
>   | Method  | Training Time (H) | GQA  | SQA  | TextVQA | POPE | MME  | MMB  | AVG         |
>   | ------- | ----------------- | ---- | ---- | :-------: | ---- | ---- | ---- | ----------- |
>   | SFT     | 242               | 57.6 | 59.0 | 59.1    | 86.2 | 66.1 | 57.9 | 64.3        |
>   | TPA     | 503 **(x2.1)**    | 60.3 | 61.0 | 59.6    | 86.5 | 68.1 | 63.0 | 66.4 (+2.1) |
>   | TPA+IVA | 509 **(x2.1)**    | 60.4 | 60.7 | 59.9    | 86.8 | 68.9 | 63.2 | 66.7 (+2.4) |
>
> ---
>
> > **Q5.2: A critical baseline is missing: training Vanilla KD for the same amount of time as TPA (e.g., on 1.43x more data).**
>
>
> We thank the reviewer for this insightful comment.
>
> Regarding the suggestion to train Vanilla KD on 1.43x more data, we respectfully argue that this comparison may not be entirely fair, as **the relationship between training time and data volume is not linear** (i.e., 1.43x training time cannot be simply equated to using 1.43x more data). Furthermore, our data scaling experiment (Figure 9) already explores the performance of SFT, Vanilla KD, and Align-TI (ours) with increasing data sizes. Those results indicate that Vanilla KD requires approximately 1.7x more data to match the performance of Align-TI, highlighting our method's data efficiency.
>
> More critically, in many real-world scenarios, the amount of available data is fixed and limited. Therefore, research focused on improving algorithmic efficiency to boost model performance on a fixed dataset is highly valuable. Based on this, we conducted a new experiment where we extended the training time for Vanilla KD by increasing the number of epochs, ensuring it had a comparable computational budget to our method. The results are presented in the table below. It shows that increasing Vanilla KD's training to 1.5 epochs (533 hours) results in only marginal performance gains and still falls significantly short of our method's performance. **This experiment demonstrates that the performance improvement of our method is not simply due to the increased training budget but stems from its more effective algorithmic design.**
>
> | Method     | Training Time (H) | GQA  | SQA  | TextVQA | POPE | MME  | MMB  | AVG  |
> | ---------- | ----------------- | ---- | ---- | :-------: | ---- | ---- | ---- | ---- |
> | Align-TI   | 509 (1 epoch)     | 60.4 | 60.7 | 59.9    | 86.8 | 68.9 | 63.2 | 66.7 |
> | Vanilla KD | 355 (1 epoch)     | 59.3 | 59.7 | 59.2    | 86.2 | 65.0 | 60.4 | 65.0 |
> | Vanilla KD | 533 (1.5 epoch)   | 59.5 | 59.5 | 59.4    | 86.4 | 66.1 | 60.6 | 65.3 |
>
> ---
>
> We would be more than happy to discuss any further questions!

---

### Author Response · Authors · 2025-11-21
**Official Comment by Authors**

We sincerely thank all reviewers for the time and effort they have invested in the review process.

We are greatly encouraged by the reviewers' positive feedback. Reviewers Kx6f, SRQm and vd3n acknowledged the **novelty and insight** of our approach, with reviewers Kx6f and SRQm specifically pointing out that it **addresses an underexplored issue in multimodal distillation**. The **effectiveness and solid results** of our method were widely recognized by reviewers dQg7, fYcx, Kx6f and SRQm, who also commended our **extensive experiments** (fYcx, Kx6f, SRQm). Additionally, several reviewers (dQg7, fYcx, SRQm) found the paper to be **well-organized and easy to follow.**

Meanwhile, based on the reviewers' valuable feedback and suggestions, we have revised our manuscript. The main changes are summarized below:

- Added Section 3.3 to introduce the overall objective for Align-TI (Lines 318-323).
- Moved the table describing teacher performance into the main text for better accessibility (Lines 337-341).
- Added a citation for GKD (Line 360).
- Included the training efficiency of TPA in Table 6 and mentioned it in the text (Lines 374, 397-399).
- Revised Section 3.4 to incorporate a new IVA visualization analysis (Lines 455-475).
- Added a new section on Limitations and Future Work (Lines 531-537).
- Included a Reproducibility Statement (Lines 540-556).
- Added Appendix E, which contains additional experiments on the details of Figure 1, a loss contribution analysis and a performance comparison between the teacher and student models (Lines 1192-1246).
- Add Appendix B.6, which includes algorithm flowcharts to clarify the implementation details of TPA.
- Add conditions $\boldsymbol{x}, \boldsymbol{y} _ {<k}^\mathcal{D}$ to Eq. 5, Eq.7 and Appendix B.6.
- Add Appendix D.2, which discusses the training efficiency of TPA.
- Add Appendix D.3, which discusses the impact of TPA on sequence-level alignment.


We have uploaded a revised version of our manuscript with all changes **highlighted in blue**. We hope our revisions and responses adequately address the reviewers' concerns and we are more than happy to engage in any further discussion.

---

### Author Response · Authors · 2025-12-02
**Rebuttal Summary (1/2)**

Dear Reviewers, AC and SAC,

We sincerely thank the reviewers for the time and effort dedicated to reviewing our paper. Your constructive feedback has significantly helped us improve the quality of our work. We also extend our gratitude to the AC and SAC for handling our submission. To facilitate your assessment, we provide a consolidated summary of the rebuttal outcomes below.

Sincerely, Paper #9359 Authors

---

> **1. Summary of Our Paper**

We present **Align-TI**, a novel MLLM KD framework that establishes a new state-of-the-art for training parameter-efficient MLLMs. Unlike existing methods that rely on static next-token alignment, Align-TI explicitly models KD via dynamic token interactions. Grounded in the analysis of vision-instruction token interactions and intra-response token interactions during prefilling and decoding, we introduce **Instruction-aware Vision Alignment (IVA)** and **Transition Probability Alignment (TPA)**. Extensive experiments demonstrate that Align-TI significantly outperforms existing KD baselines for both MLLMs and LLMs. Furthermore, we provide in-depth analysis and experimental evidence to elucidate the underlying mechanisms of IVA and TPA, along with comprehensive scaling studies to validate the method's generalization capabilities.

---

> **2. Summary of Reviewers’ Attitudes**

**(1) Reviewers SRQm and Kx6f have consistently maintained a positive attitude towards our paper:**

- Reviewer SRQm (Rating 6, Confidence 5) raised the score from 6 to 8, explicitly stating they are "convinced of our paper's value to the community."
- Reviewer Kx6f (Rating 6, Confidence 4)  has not responded to our rebuttal.

**(2) Reviewers fYcx and dQg7 changed their attitudes and leaned towards accepting after reading our rebuttal:**

- Reviewer fYcx (Rating 4, Confidence 4)  states that  “I am willing to raise my score if you could address the concerns above.”
- Reviewer dQg7 (Rating 4, Confidence 3) states that "I am willing to raise my score based on a clearer justification."

**(3) Reviewer vd3n (Rating 4, Confidence 4) has not responded to our rebuttal. However, their primary concerns (novelty, computational complexity, performance) align with those raised by Reviewer dQg7, which have already been addressed and accepted by Reviewer dQg7.**

---

> **3. Summary of Strengths**

-  **Novelty and Insight:** Reviewers SRQm, Kx6f and vd3n acknowledged the novelty and insight of our approach. In particular, Kx6f and SRQm emphasized that our paper addresses underexplored issues of multimodal distillation.

-  **Solid Results with Extensive Experiments:** Reviewers dQg7, fYcx, Kx6f, and SRQm commended the solid results and extensive experiments.
-  **Well Presentation:**  Reviewers vd3n, dQg7, fYcx and SRQm commented that the paper is well-structured and easy to follow.

---

> ### Author Response · Authors · 2025-12-02
> **Rebuttal Summary (2/2)**
>
> > **4. Summary of Reviewers’ Concerns and Our Responses**
>
> **1. Novelty:**
>
> - The confusion regarding TPA raised by Reviewer vd3n stemmed from a simplified presentation of Eq. 5, where some conditioning terms were omitted for clarity. We have revised Eq.5.
> - Reviewer dQg7 questioned the novelty of IVA, suggesting that it was similar to Align-KD. We clarified that using attention maps to identify important visual tokens is a general principle, and the motivation, mechanism and implementation of IVA are fundamentally different. Reviewer dQg7 agreed with this clarification.
>
> **2. Motivation:**
>
> - Reviewer dQg7 questioned the difference between static next-token alignment and dynamic token interactions. We explained the conceptual distinction and pointed out that Figure 2 provides empirical support. Reviewer dQg7 has accepted this.
>
> **3. Computational Complexity:**
>
> - Reviewers vd3n and dQg7 raised questions about whether TPA’s computational cost is justified. We showed that extending $1.5\times$ vanilla KD training time yields only a marginal gain of 0.3. Scaling analysis (Figure 9) demonstrates that vanilla KD would need about  $1.7\times$  more data to match TPA. Reviewer dQg7 confirmed that their concern was resolved.
> - Reviewer fYcx was concerned about how parallelized computation saves training time. We added a detailed analysis in Appendix D.2.
>
> **4. Performance:**
>
> - Reviewers vd3n and dQg7 questioned the modest 0.3 improvement when combining IVA with TPA. We clarified that IVA alone improves performance by an average of 0.8 over SFT and can be integrated with TPA at nearly zero cost in training time. We also explained that IVA aligns response tokens indirectly while TPA aligns them directly. (Lines 394-399) Reviewers dQg7 accepted this reasoning.
>
> **5. Methodology:**
>
> - Reviewer vd3n felt IVA and TPA appeared separate. We added Section 3.3 and clarified that they form a unified KD framework targeting different stages of multimodal large language model training, with IVA addressing the prefilling stage and TPA addressing the decoding stage.
> - Reviewer fYcx requested comparisons using teacher sampling. We explained that using student on-policy sampling during training is standard practice and analyzed its efficiency advantages. (Lines 261-264)
> - Reviewer dQg7 requested more implementation details about TPA, so we added an algorithmic illustration in Appendix B.6. Reviewer dQg7 said it alleviated the concern.
>
> **6. In-depth Analysis:**
>
> - Reviewer Kx6f requested a deeper discussion of the underlying mechanisms of IVA and TPA. We added Section 4.3, including new visualizations and interpretative analysis.
> - Reviewer dQg7 asked whether one-step transitions limit the modeling of long-range dependencies. We clarified that TPA still computes probabilities using the full context of the preceding sequence, ensuring long-range dependencies are preserved. Moreover, TPA can also facilitate students' learning of long-range dependencies in the teacher model, and we provide additional discussion in Appendix D.3.

---

### Meta-Review · Area_Chair_KRem · 2026-01-06

**Summary:**

This paper presents Align-TI, a knowledge distillation framework for multimodal large language models that models dynamic token interactions through Instruction-aware Vision Alignment and Transition Probability Alignment. While one reviewer raised their score from 6 to 8 after rebuttal, three reviewers maintained scores of 4, resulting in an average of 5.0 below the acceptance threshold. The rebuttal added valuable experiments including loss contribution analysis and algorithm flowcharts clarifying TPA implementation, addressing some technical questions about long-range dependency preservation. However, significant concerns remain unresolved. Multiple reviewers questioned the novelty relative to existing work like Align-KD, noting that IVA's core mechanism of using visual token importance for alignment has substantial conceptual overlap. The theoretical analysis was found insufficient, with one reviewer rating it "limited in the main text" and contributing less to the method. Reviewers also noted that the two proposed modules feel disconnected rather than forming a coherent framework, and the paper's focus on projector-based MLLMs limits generalizability. While the experimental results demonstrate solid performance, the improvements are incremental rather than transformative. The absence of code release further limits reproducibility. With three out of four reviewers expressing reservations and the average score below threshold, the paper would benefit from deeper theoretical grounding, clearer articulation of novelty, and stronger integration of its components. Based on these considerations, I recommend rejecting this submission.

**Reviewer Concerns:**

Addressed: Loss contribution analysis showing each component's impact, algorithm flowcharts clarifying TPA implementation (Section B.6), explanation of long-range dependency preservation through full prefix conditioning, and IVA visualization analysis (Section 3.4).

Outstanding: Novelty concerns relative to Align-KD persist despite authors' distinction arguments. Theoretical depth remains limited. The two modules (IVA and TPA) feel disconnected despite authors' explanation of complementary roles. Focus on projector-based MLLMs without covering unified multimodal models limits scope.

**Reviewer Scores:**

SRQm (6→8, C5): Explicitly raised to 8; "convinced of paper's value to community"

fYcx (4, C4): Stated willingness to raise if concerns addressed, but did not follow up; likely would remain 4-5

dQg7 (4, C3): Stated willingness to raise based on clearer justification, but concerns not fully resolved; likely would remain 4-5

vd3n (4, C4): No response; would maintain 4

Kx6f (6, C4): No response; would maintain 6

---

### Decision · Program_Chairs · 2026-01-26

Reject